# Heiner Syndrome and Milk Hypersensitivity: An Updated Overview on the Current Evidence

**DOI:** 10.3390/nu13051710

**Published:** 2021-05-18

**Authors:** Stefania Arasi, Carla Mastrorilli, Luca Pecoraro, Mattia Giovannini, Francesca Mori, Simona Barni, Lucia Caminiti, Riccardo Castagnoli, Lucia Liotti, Francesca Saretta, Gian Luigi Marseglia, Elio Novembre

**Affiliations:** 1Translational Research in Pediatric Specialities Area, Division of Allergy, Bambino Gesù Children’s Hospital, IRCCS, Piazza Sant’Onofrio, 4, 00165 Rome, Italy; 2Pediatric Unit and Emergency, University Hospital Consortium Corporation Polyclinic of Bari, Pediatric Hospital Giovanni XXIII, 70126 Bari, Italy; carla.mastrorilli@icloud.com; 3Department of Medicine, University of Verona, 37129 Verona, Italy; luca.pecoraro@assst-mantova.it; 4Pediatric Unit, ASST Mantua, 46100 Mantua, Italy; 5Allergy Unit, Department of Pediatrics, Meyer Children’s University Hospital, 50139 Florence, Italy; mattia.giovannini@unifi.it (M.G.); francesca.mori@meyer.it (F.M.); simona.barni@meyer.it (S.B.); 6Department of Human Pathology in Adult and Development Age “Gaetano Barresi”, Allergy Unit, Department of Pediatrics, AOU Policlinico Gaetano Martino, 98158 Messina, Italy; lucycaminiti@yahoo.it; 7Department of Pediatrics, Pediatric Clinic, Fondazione IRCCS Policlinico San Matteo, University of Pavia, 27100 Pavia, Italy; riccardo.castagnoli@yahoo.it (R.C.); gl.marseglia@smatteo.pv.it (G.L.M.); 8Department of Pediatrics, AOU Ospedali Riuniti Ancona, Presidio Ospedaliero di Alta Specializzazione “G. Salesi”, 60126 Ancona, Italy; lucia.liotti@ospedaliriuniti.marche.it; 9Pediatric Department, Latisana-Palmanova Hospital, Azienda Sanitaria Universitaria Friuli Centrale, 33100 Udine, Italy; francescasaretta@gmail.com; 10Department of Health Science, University of Florence, 70126 Florence, Italy; elio.novembre@meyer.it

**Keywords:** allergy, anemia, cow’s milk, children, immunology, non-IgE-mediated food allergy, pneumonia, pulmonary hemosiderosis, pulmonary infiltrates

## Abstract

Infants affected by Heiner syndrome (HS) display chronic upper or lower respiratory tract infections, including otitis media or pneumonia. Clinically, gastrointestinal signs and symptoms, anemia, recurrent fever and failure to thrive can be also present. Chest X-rays can show patchy infiltrates miming pneumonia. Clinical manifestations usually disappear after a milk-free diet. The pathogenetic mechanism underlying HS remains unexplained, but the formation of immune complexes and the cell-mediated reaction have been proposed. Patients usually outgrow this hypersensitivity within a few years. The aim of this review is to provide an updated overview on the current evidence on HS in children, with a critical approach on the still undefined points of this interesting disease. Finally, we propose the first structured diagnostic approach for HS.

## 1. Introduction

Heiner syndrome (HS) is a rare food-induced hypersensitivity disease characterized by chronic respiratory symptoms with X-ray (XR) infiltrates, and the resolution of signs and symptoms after the removal of milk proteins. Other clinical manifestations include poor growth, gastrointestinal signs and symptoms, iron deficiency anemia and pulmonary hemosiderosis (PH). Precipitins to cow’s milk (CM) were also considered a useful aid in recognizing hypersensitivity to CM [1]. The literature concerning HS is restricted to a few case reports or series, although the disease has always been described at infant or pediatric age [2]. The definition of the disease is lacking and the diagnosis is often delayed, since its presentation is uncommon with dissimilar manifestations. In the present review we aimed at presenting clinical, diagnostic and therapeutic characteristics of HS, starting from current evidence.

## 2. Search Methodology and Results

We carried out a non-systematic review including the most relevant studies on “Heiner Syndrome” (HS) present on databases including PubMed (https://www.ncbi.nlm.nih.gov/pubmed/ accessed on 26 March 2021), MEDLINE, The Cochrane Library, from their inception to 26 March 2021. The searched terms were “Heiner Syndrome” [all fields]; “pulmonary hemosiderosis” and “children” [all fields]; “pulmonary hemosiderosis” and “cow’s milk” [all fields]; “pulmonary hemosiderosis” and “hypersensitivity” [all fields]. We found 16 studies. They were all clinical cases or consecutive case series, involving an overall pool of 61 patients. Findings were summarized narratively below for each study as well as in Table 1.

In order to better stratify the level of evidence for the diagnosis, we are herein proposing the first structured diagnostic criteria for HS to our best knowledge. This diagnostic approach consists of the following criteria:(A)Pulmonary symptoms and XR infiltrates or pulmonary hemosiderosis (PH);(B)Resolution after milk removal;(C)Recurrence after milk reintroduction.

HS diagnosis (HSD): (A) + (B) = probable disease; (A) + (B) + (C) = convincing disease (Figure 1).

Milk precipitins have been evaluated only in some studies, mainly in the oldest ones. Based on current data and specifically due to the heterogeneity in the methodologies applied for laboratory tests, we decided to exclude laboratory parameters from the diagnostic approach.

(1)**Heiner et al., 1960** [3]—In 1960, Heiner [3] (from who the name of the syndrome originates) first reported a chronic respiratory disease associated with multiple CM precipitins in the sera of seven children aged 6 weeks to 17 months. All the patients presented a chronic respiratory disease with maximal severity of clinical manifestations at the time of introduction of raw CM in the diet associated with the other signs and symptoms that define the syndrome, mostly iron deficiency anemia, gastrointestinal signs and symptoms, poor growth and PH documented by gastric or bronchial aspirates. Six out of seven patients overcame their disease by changing the milk content (i.e., milk processing or exclusion of milk proteins) in their diet, by using evaporated or boiled milk (n = 2), extensively hydrolyzed casein formula (n = 1) or soymilk (n = 3). Overall, the milk protein avoidance resulted in the complete disappearance of clinical manifestations and remission of the hematologic status. One patient spontaneously overcame the disease without dietary restriction between 2.5 to 3.5 years (y) of age. When milk was reintroduced after avoidance, two out of six showed a clinical and imaging relapse, and four became tolerant or partially tolerant to milk after intervals without clinical manifestations on a restricted diet ranging from three to six months in duration.**Comment:** The first description of HS.**HSD:** Convincing in two cases; probable in four patients.(2)**Holland et al., 1962** [4]—Stimulated by Heiner’s observations, Holland et al. examined serum specimens from 1618 infants and children with the same technique, finding precipitins in 87 of them [4]. Patients of this population showed signs and symptoms suggestive of the syndrome but also different clinical manifestations, such as isolated upper respiratory diseases, hepatosplenomegaly and congenital heart diseases. Only 17 patients were reported to show respiratory signs and symptoms. No data on X-rays were reported. Because of the heterogeneity of the clinical manifestations and the limited number of diagnostic exams in this population, we selected 22/87 patients who improved during the CM diet period. No attempt of reintroduction was performed.**Comment:** We extrapolated 22 cases with suspected HS from a large and heterogeneous cohort.**HSD:** Nobody with probable or convincing clinical criteria.(3)**Chang et al., 1969** [5]—Chang described the clinical case of a 9-month-old girl admitted to the hospital for failure to thrive (FTT), anemia and chronic recurrent lung disease starting in the first weeks of age [5]. She underwent a lengthy diagnostic process, until the finding of milk precipitins suggested an HS diagnosis. The patient was then placed on a soymilk diet with clinical resolution. Flare-up signs and symptoms and radiological relapse due to the poor adherence to the diet are mentioned. However, controlled milk reintroduction was not performed.**Comment**: Single case report. No detailed data are reported about the follow-up.**HSD**: Probable.(4)**Archer, 1971** [6]—Archer reported the clinical case of a 13-month-old girl with a severe heart failure based on a profound iron deficiency anemia and idiopathic PH, diagnosed by needle biopsy [6]. All of the immunological tests performed in order to investigate a CM sensitization, including serum precipitins, skin prick test (SPT) and immunoglobulin, were negative. Notwithstanding, a milk-free diet was commenced with a good clinical and radiological response. The first relapse during her first week with a childminder was probably due to the inadvertent administration of CM.**Comment**: The girl was admitted one year before with some symptoms and treated successfully with antibiotics. Results of CM reintroduction doubtful.**HSD:** Probable.(5)**Boat et al., 1975** [7]—In this study, 6 children with high titers of milk precipitins were identified by screening the sera of 160 children with idiopathic chronic lung disease associated with typical manifestations of milk-induced PH [7]. Elimination of CM from the diet, symptomatic therapy and adenoidectomy (when indicated) resulted in improvement in six out of six patients.**Comment:** Even if six out of six patients recovered in 5–21 days after milk removal, only one was rechallenged (Patient C) and one (Patient B) developed pneumonia within six months upon CM reintroduction.**HSD:** Convincing in one case; probable in five patients.(6)**Stafford et al., 1977** [8]—Nine patients with respiratory signs and symptoms and milk precipitins were enrolled in this study in order to elucidate the immunopathologic mechanisms involved in milk-induced PH. No demonstration of a unique immunologic mechanism associated with milk-induced PH in the patients studied [8]. No clinical data about CM withdrawal were reported.**Comment:** Focus on immunological patho-mechanisms with poor clinical description of the enrolled participants.**HSD:** Nobody with probable or convincing criteria.(7)**Fossati, 1992** [9]—A 7-year-old girl was admitted to hospital because of anemia, and a PH was diagnosed [9]. Precipitating antibodies were also found. A marked improvement of clinical manifestations and XR results were found after removal of CM from the diet.**Comment**: Single clinical case report. No data on reintroduction.**HSD:** Probable.(8)**Torres et al., 1996** [10]—This study provides an interesting immunological overview on the HS based on data from a single clinical case of a girl [10]. Specifically, the authors speculate that an inflammatory response occurred after CM intake in the presented clinical case. A 5-day-old female newborn was admitted in an emergency department because of a severe bradycardia due to a myocarditis associated with assessed PH and anemia. Although the total specific IgE and specific IgG to milk proteins were below the detection limits, the patient underwent two oral food challenges (OFC). During the in vivo tests, hematocrit, histamine, tryptase and ECP (eosinophil cationic protein) in blood and BAL, and N-methyl-histamine (NMH) in urine were measured before and at multiple times during the administration of standard formula (first OFC) and non-milk enteral nutrition (second OFC). When the girl was fed with CM, a remarkable increment of all the tested inflammatory mediators was reported; conversely, the hemoglobin level dropped significantly. Then, the same OFC was done with non-milk enteral nutrition with any variation being registered. These data are unfortunately not supported by clinical and radiologic information. Challenge.**Comment:** Unique case report of PH in which milk OFC induced an increase of inflammatory mediators suggesting a T cell-mediated pathogenesis of HS.**HSD:** Probable.(9)**Moissidis et al., 2005** [11]—Moissidis et al. reviewed eight cases of children affected by upper respiratory tract symptoms [11]. All cases presented radiological imaging with pulmonary infiltrates, and one had HP (defined as iron-laden macrophages in the bronchoalveolar lavage, gastric washing and open lung biopsy). Seven out of eight had gastrointestinal symptoms. High titers of precipitating antibodies to CM proteins were demonstrated in six out of six patients studied. However, HS was confirmed by the improvement of the clinical and radiological findings after a CM-free diet and relapse when a reintroduction was attempted in three out of six cases.**Comment:** The most detailed paper on the topic. However, cases were evaluated at different times and under different circumstances; therefore, specific data were not available for each patient.**HSD**: Convincing in three cases; probable in five patients.(10)**Sigua et al., 2013** [12]—A 12-month-old boy with multifocal pneumonia that was refractory to protracted antibiotic treatment was suspected to suffer from HS [12]. The clinical history showed that the boy underwent a milk-free diet from the first to the tenth month of age because of suspected non-IgE-mediated CM non-bloody diarrhea. HS appeared at CM reintroduction. Serum-precipitating IgG antibodies to all nine CM protein fractions tested were strongly positive. He underwent a strict soy-based diet from 12 months of age with prompt clinical remission and complete resolution of the previously identified pulmonary opacities at a chest X-ray performed at 14 months.**Comment:** A single patient with HS after a previous history of non-IgE-mediated CM gastrointestinal symptoms.**HSD:** Probable.(11)**Yavuz et al., 2014** [13]—A 3-year-old boy was referred to the emergency service with respiratory distress and hemoptysis [13]. Because of iron deficiency anemia, a BAL cytological examination was performed in order to confirm a PH. Precipitins were not determined. The patient overcame the disease through a CM avoidance diet. However, a low dose of both prednisolone and azathioprine was also prescribed. Furthermore, authors described that the patient in the next five years had many relapses because of failure to receive the prescribed medications and poor adherence to the diet. Moreover, during a hemoptysis attack, he showed new symptoms, such as edema, hematuria and hypertension. On this occasion, rapidly progressive glomerulonephritis was diagnosed on the basis of the histopathological findings and treated with a combination of cyclophosphamide and methylprednisolone.**Comment:** In this case report, an elimination diet and drugs were administrated together for an extended period of time, and during the follow-up the compliance was scarce. Therefore, it is difficult to differentiate the effects of each treatment and the actual cause-effect relationship.**HSD:** Probable.(12)**Mourad et al., 2015** [14]—The clinical case of a 17-month-old boy with idiopathic PH was described by Mourad et al. [14]. BAL demonstrated an abundance of fresh red blood cells and iron-laden macrophages. The CM-specific IgE level was only slightly elevated (1.42 IU/mL). IgG antibody levels to CM proteins were markedly elevated. In spite of the severity of the clinical conditions (i.e., severe anemia and respiratory failure with acidosis), the strict CM-free diet allowed the boy to overcome the disease. Hydrocortisone was also administrated, but it is not clear when it was introduced and for how long. A relapse was reported because the mother, while on raw CM avoidance, started feeding the patient with baked CM products.**Comment**: Single clinical case report. Hydrocortisone was also used. Controlled reintroduction not performed.**HSD:** Probable.(13)**Alsukhon et al., 2017** [15]—A 2-month-old male with recurrent diarrhea and FTT had persistent cough, tachypnea and high inflammatory markers despite antibiotic therapy for pneumonia [15]. An amino acid-based formula gave improvement in inflammation and respiratory function.**Comment:** Single case report. Milk reintroduction not performed.**HSD:** Probable.(14)**Ojuawo et al., 2019** [16]—Ojawo et al. described the clinical case of a 16-week-old boy with FTT, dyspnea and anemia who acceded to the emergency department in Nigeria [16]. Neither antibiotic treatment nor sodium citrate, administered for the suspicion of a renal tubular acidosis, modified his condition. Diagnosis of HS was based on the constellation of clinical features, XR results, and subsequent resolution after stopping CM. Parents on a follow-up visit reported occasional cough and rhinitis whenever CM was reintroduced.**Comment:** Single case report. No controlled CM reintroduction reported.**HSD:** Probable.(15)**Koc et al., 2019** [17]—A 6-month-old infant with massive hemoptysis, hematemesis and deep anemia was treated for bronchopneumonia four times [17]. When he was admitted to the emergency department, both chest-X ray and computerized tomography documented many lung opacities and hemosiderin-laden macrophages were found in the patient’s fasting stomach fluid examination, confirming the diagnosis of PH. The boy was discharged with a CM-free diet, with complete clinical and radiological recovery.**Comment:** No laboratory data were reported, and no milk reintroduction test was reported.**HSD**: Probable.(16)**Liu et al., 2020** [18]—Liu et al. described a 4-month-old boy with a chronic pulmonary syndrome whose main presenting symptom was a persistent hematochezia since the tenth day of life [18]. Gastrointestinal endoscopic biopsy showed granulation tissue infiltrated by acute and chronic inflammatory cells, including some eosinophils. Additionally, in this case, the improvement of both clinical and radiologic findings after the elimination of milk suggested the diagnosis of HS. In addition to the CM elimination diet, the patient was treated with methylprednisolone (1 mg/kg) and montelukast.**Comment:** Single case report. No milk reintroduction test reported.**HSD:** Probable.

## 3. Discussion

To the best of our knowledge, the present article represents the first review on this rare disease. The data shown suggest a critical approach to the disorder (Table 1).

### 3.1. Age at Onset

The clinical onset of the disease has been described typically by the age of 1 month to 48 months, but it can develop even during the first 5 days of life, as reported by Torres et al. [10]. However, it can also appear later (the oldest patient was 5 years old) [4]. Nevertheless, there was a frequent delay in diagnosing this disease, due to its various modes of presentation and lack of standardized diagnostic criteria. The past medical history of affected children was always unremarkable. A family history of allergic disorders was often present. Differently from immediate-onset IgE-mediated CM allergy, HS did not display signs and symptoms before several days or weeks after CM consumption.

### 3.2. Etiology

Although HS is more likely to be induced by homogenized CM, the disease also may occur in some infants fed with CM-derived formula. Furthermore, it has been speculated that it can be related to non-IgE-mediated allergy to food proteins differently from CM at an older age (e.g., soy, egg, pork, wheat and peanut) [14,19]. In this context, a single case of PH hemosiderosis due to buckwheat has also been reported [20].

### 3.3. Clinical Characteristics

Respiratory features of the disease included persistent cough, dyspnea, tachypnea, wheezing, occasional sputum production and rales. The peculiarity of pneumonia in these case series was the refractoriness of antibiotic treatments. Of note, in most cases, the additional administration of anti-inflammatory drugs probably might have resolved hypersensitivity pneumonia or idiopathic PH (IPH). The most commonly described systemic clinical manifestations were intermittent fever, progressive anorexia and FTT. Inflammatory markers were usually found to be high. Eosinophilia and severe iron deficiency anemia were frequently described at blood count examination. Gastrointestinal manifestations were reported in about half of the patients and included frequent vomiting or diarrhea. Rarely, lymph node hypertrophy with hepatomegaly, splenomegaly and hypertrophied tonsils or adenoids were labeled [7]. Noticeably, lymphonodular hyperplasia in biopsy was found in a child with HS-manifesting hematochezia [18].

Clinically, the disease can be complicated with cardiopulmonary involvement, such as alveolar hypoventilation, massive acute PH, pulmonary hypertension and *cor pulmonale*, or nephrological ones, such as crescentic glomerulonephritis [8,13]. These characteristics contributed drastically to morbidity and emerged in situations of overdue diagnosis and management. In particular, a delayed manifestation of the disease is episodic hemoptysis, which may represent a PH with repeated episodes of intra-alveolar bleeding, hemosiderin deposition in alveolar macrophages, followed by the development of pulmonary fibrosis and severe anemia [8]. PH may occur as a primary disease of the lung (also called IPH) or secondary to cardiac diseases, bleeding disorders, collagen-vascular diseases or systemic vasculitis. IPH, if not treated, leads to progressive pulmonary fibrosis and may be lethal [21].

### 3.4. Pathogenesis and Immunological Implications

The exact mechanism that triggers HS is not fully understood. Feasibly, the formation of immune complexes is strongly suspected (Gell and Coombs type III reaction) and the cell-mediated reaction (Gell and Coombs type IV reaction) may contribute to the development of this challenging disease.

Some cases showed positive skin tests [3,7], high serum total IgE levels [7], high milk-specific IgE antibodies [13,14] or circulating immune complexes [3]. A significant increase of histamine and ECP in BAL several hours after a milk OFC was reported [10]. In one case report, deposits of immunoglobulins, complement, fibrin and milk protein antigens diffusely scattered were described on immunofluorescence studies of lung tissue biopsies [2]. It is probable that a cause concurring to HS is the aspiration of milk, in particular among patients with an uncoordinated swallowing mechanism, tracheal/esophageal anomaly or gastroesophageal reflux. However, in the paper of Boat et al., this condition was ruled out [7]. Concerning the data on delayed immunity, in some cases a delayed skin test response to intradermal test [3,7] or a lymphocyte response [8] was reported. Other authors postulated that milk antigens might trigger an immune complex reaction resulting in multiorgan abnormalities, such as pulmonary, gastrointestinal and renal ones. In fact, pulmonary and gastrointestinal signs and symptoms were frequently associated [11] and granular immuno-deposits have been demonstrated along the glomerular basement membrane in a child with crescentic glomerulonephritis and PH [13].

Most studies, mainly the oldest ones, characteristically found high titers of precipitins (likely immunoglobulins of class G) against bovine milk proteins in the patients’ sera, by using the Ouchterlony double-immunodiffusion technique [4,5,7,11]. However, it is not sufficiently explicable why some children develop precipitating antibodies to ingested protein and other children do not. Moreover, it is not known if these precipitins play a causative role in the disease. Children with precipitins usually have an increased incidence of recurrent respiratory tract diseases, anemia and hepatosplenomegalia. However, precipitating IgG antibodies to milk are not pathognomonic of the disease, since they have been found among around 1% of healthy children in the absence of clinical manifestations [4], and in 4% to 6% of children with chronic disorders, including celiac disease, cystic fibrosis, IgA deficiency, Down’s syndrome, Wisckott–Aldrich syndrome and Hurler’s syndrome [7]. Additionally, methods used for detecting precipitins are obsolete and reports on them are timeworn. Therefore, the role of these antibodies should be critically considered.

In other more recent reports, high values of specific IgG to milk proteins were found using an immunoenzimatic technique [14]. In one case report [15] CM IgG4 was found to be elevated. Even in these cases, the role of this specific CM IgG is not clear.

### 3.5. Pulmonary Modifications and Immunofluorescence Studies

Chest roentgenograms displayed variable patchy and transient infiltrates, frequently associated with areas of atelectasis, consolidation, reticular opacities, pleural thickening or hilar lymphadenopathy. The lung biopsy obtained in patients who had hemoptysis showed an abnormal accumulation of hemosiderin in the lungs, which resulted in alveolar hemorrhage or PH [2]. Among patients with anemia and hemoptysis, PH was verified by the demonstration of iron-laden macrophages by using Prussian Blue staining of bronchial aspirates or morning gastric washes [8].

### 3.6. Diagnostic Criteria

As described in detail above, in order to better stratify the level of evidence for the diagnosis, we are herein proposing the first structured diagnostic approach for the diagnosis of HS to the best of our knowledge. We auspicate that this approach may allow clinicians to stratify patients with a clinical history consistent with the suspicion of HS in probable (criteria A + B) or convincing (criteria A + B + C) HS (Figure 1). According to our criteria, only 6 out of 61 patients had a convincing clinical diagnosis of HS, in 25 patients the clinical diagnosis was probable and in the others the HS diagnosis was doubtful (Table 1). Due to the heterogeneity in the methodologies applied for laboratory tests and missing reporting/lack of data, we decided to exclude laboratory parameters from the diagnostic criteria. We suggest in the future that milk-specific IgG tests with current diagnostic methods (i.e., immunoenzymatic), in the case of suspected HS, could be studied. Nevertheless, further points remain questionable in our proposed diagnostic approach. First, some cases of occurrence of signs and symptoms during exclusive maternal breastfeeding were reported (e.g., in three out of seven cases in Heiner et al. [3], leading to the question of whether or not minimal quantities of CM passing in breast milk are capable of inducing a clinical response through an IgG-mediated mechanism in the infant). As a second observation, one case of resolution of signs and symptoms without dietary restriction has been described (one out of seven patients from Heiner et al.). Again, a real, controlled OFC of milk was reported only in 6 cases [3,7,11] and a recurrence of clinical manifestations was reported in a further 6 cases (although without details on exact timing of the symptoms’ onset) based only on clinical history [5,6,7,13,14,16], for a total of 12 cases. Moreover, in some cases [13,14] pharmacological therapy was associated to the CM elimination diet, making it difficult to differentiate the effect of each single treatment. Spontaneous resolution of signs and symptoms also occurred in some cases of HS [1]. Furthermore, some studies did not report on the follow-up and specifically on the outcome of any CM reintroduction. In conclusion, even if in a few cases a convincing diagnosis can be made using specific criteria, certainty is lacking due to the incomplete clinical and imaging monitoring of the OFC and the lack of control cases.

### 3.7. Differential Diagnosis

In the differential diagnosis (DD) bronchial asthma, chronic aspiration, acute and chronic lower respiratory tract infections, including fungal ones, cystic fibrosis, foreign body, hypersensitivity pneumonitis, bronchopulmonary aspergillosis, secondary PH and IPH should be considered [11]. Cystic fibrosis essentially can be excluded by a normal chloride level on the sweat test and second line tests if there is a strong suspicion. Infectious causes should be ruled out with tuberculin skin test and available microbial tests, and empiric antibiotic treatments characteristically are inefficacious. Thereafter, a vasculitis or autoimmune disorder can be considered. The characteristic pulmonary hemorrhage attacks cannot be enlightened by modest bronchial asthma. The lack of chronic inhalant exposure and BAL examination can rule out hypersensitivity pneumonitis. Bronchopulmonary aspergillosis is unlikely if skin reactivity to *Aspergillus* antigens is negative or precipitating serum antibodies to *A. fumigatus* are absent. No hemorrhagic focus nor foreign bodies can be found on bronchoscopy. In particular, DD should consider IPH, that includes the classic triad hemoptysis, radiologic lung infiltrate and iron deficiency anemia, having a more severe course and prognosis [22]. Moreover, IPH occurs in older children, and it is rarely associated with gastrointestinal symptoms [23] (Table 2).

### 3.8. Natural History

The disease’s effects were reversible by stopping CM consumption. In fact, signs and symptoms could last within a time range of 5 to 21 days after the CM withdrawal. Before CM reintroduction, some children showed spontaneous tolerance to pasteurized or boiled CM [4]. Reoccurrence of clinical manifestations was reported with CM reintroduction [3,5,6,10,11,14]. However, it is believed that patients usually definitively outgrow this hypersensitivity, and they can tolerate CM within a few years [11,24,25].

### 3.9. Treatment

Clinical improvement with strict elimination of CM proteins sustains the diagnosis of HS. Infants may be fed by a milk substitute, such as extensively hydrolyzed protein formula, soy-based formula or synthesized free amino acid formula. Improvement of signs and symptoms occurs in few days and X-ray images in weeks. When a confirmatory CM reintroduction was performed [11], respiratory clinical manifestations also occurred in days or weeks. An early exclusion of the triggering food from the diet is crucial, since chronic PH induces pulmonary fibrosis which can be fatal [6]. However, recovery may occur also without exclusion of the culprit food (e.g., 1/7 in Heiner’s report [3]).

Even if the most striking criterion of HS is the dramatic response to the exclusion diet, initially, in some cases, appropriate treatment, e.g., bronchodilators, antihistamines, systemic or inhaled steroids and iron, may be needed. A short cycle of oral corticosteroids remains the first-line therapy for acute attacks. In more severe cases of HS, other immunomodulatory treatments may be helpful, such as hydroxychloroquine, azathioprine or cyclophosphamide. On the contrary, antibiotic therapy seems not to be useful.

## 4. Conclusions

Although HS has been described as a hypersensitivity disease due to CM, there are still pros and cons about its real existence (Table 3), and a certainty of diagnosis is lacking. A challenge test was performed in a few cases and always in an open manner. Again, signs and symptoms develop in the hours or days after milk consumption, and they disappear after the elimination diet. Precipitating IgG antibodies are an old-fashioned laboratory test reported in some patients during the disease; precipitins diminish or disappear during the elimination diet. However, they are not pathognomonic nor specific for the disease diagnosis and their pathogenetic role is still unclear. The prognosis is generally good, even if the extent of the exclusion diet necessary to reach a complete recovery is unclear. HS is a very intriguing disease, in some ways still controversial, but it is important to know and suspect this rare syndrome in any infant or young child with unexplained chronic pulmonary clinical manifestations. The diagnosis should be proved by clinical and radiologic improvements after strict CM avoidance with the recurrence of signs of symptoms and imaging features after a controlled CM reintroduction.

Considering the high clinical impact of the disease and the associated morbidity, more attention should be devoted to it, both in terms of clinical suspicion and research on the underlining patho-mechanisms with the detection of reliable biomarkers. We highlight the need for more stringent diagnostic criteria that combine both clinical manifestations and imaging features. Moreover, a follow-up evaluation with a well-designed CM OFC/regular reintroduction is of paramount importance to better understand this disease in terms of prognosis and duration. In summary, the future establishment of validated diagnostic criteria, the awareness of specific clinical manifestations and specific imaging features and the results of CM OFC will help health professionals in clinical practice to suspect the disease and to refer patients to the appropriate specialists.

## Figures and Tables

**Figure 1 nutrients-13-01710-f001:**
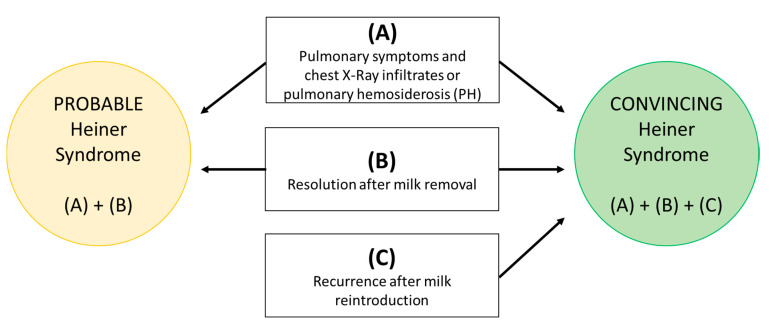
Proposed approach for Heiner syndrome diagnosis.

**Table 1 nutrients-13-01710-t001:** Characteristics of the included studies of Heiner syndrome.

	Authors, Year	Country	n. Cases	Age at Onset (Months)	Signs/Symptoms	Lung Radiologic Infiltrates	Pulmonary Hemosiderosis	Hemosiderosis Diagnosis	Milk Precipitins	Delayed Hypersensitivity	Allergic Sensitization to Milk [sIgE and/or skin prick test (SPT)]	Improvement Upon Milk Avoidance	Recurrence Upon Milk Reintroduction
1	Heiner, 1960	USA	7	1–17	chronic cough (7/7); wheezing (7/7); chronic rhinitis (7/7); frequent fever (7/7); frequent earache (6/7); diarrhea (6/7); vomiting (5/7); hemoptysis (4/7)	yes, recurrent	yes (5/7)	gastric or bronchial aspirates (4/7); pulmonary needle biopsy (1/7)	yes	intradermal test (ID) late response pos (4/7)	ID +ve (7/7)	yes (6/6)	yes (2/6)
2	Holland, 1962	USA	22	4–12	respiratory disease; failure to thrive (FTT); anemia; splenomegaly; hepatomegaly	non specified (NS)	NS	NS	yes	NS	NS	yes (22/24 on milk-free diet)	NS
3	Chang, 1969	USA	1	9	FTT ; anemia; chronic recurrent lung disease	yes	NS	NS	yes	NS	NS	yes	doubtful, based on clinical history (CH)
4	Archer, 1971	England	1	13	iron-deficiency anemia; lethargy; pallor; bloody vomit; severe cardiac failure; hemoptysis	yes	yes	needle biopsy	no	NS	SPT –ve	yes	doubtful, based on CH
5	Boat, 1975	USA	6	7–48	idiopathic chronic or recurrent pulmonary disease; upper respiratory symptoms; FTT (3/6); frequent regurgitation and watery stools (1/6). SOF (1/6): Iron deficiency (5/6); Anemia (4/6); Eosinophilia (4/6); right ventricular hypertrophy (3/6); adenoid hypertrophy (3/6)	yes (6/6)	yes (5/6)	gastric washing or bronchoalveolar lavage (BAL)	yes (6/6)	4 ID delayed response	total IgE –ve (6/6); SPT +ve (5/6)	yes (5/6); 1/6 loss of data	yes (1/6)
6	Stafford, 1977	USA	7	8–48	wheezing (5/9); chronic rhinitis (3/9); large adenoids/tonsils (4/9); anemia (4/9); gastrointestinal symptoms (4/9); eosinophilia (6/9)	yes (7/9)	yes (3/9)	gastric washing and BAL	yes (9/9)	lymphocyte response (3/3)	SPT +ve (6/9), sIgE +ve (5/8)	NS	NS
7	Fossati, 1992	Italy	1	7 years	anemia; respiratory symptoms	yes	yes	NS	yes	NS	NS	yes	NS
8	Torres, 1996	Spain	1	0 (5 days)	vomiting with blood; respiratory failure; restrictive miocardiopathy; anemia; eosinophilia	yes (chest X-Ray, CXR)	yes	BAL	NS	NS	neg	yes	symptoms not reported
9	Moissidis, 2005	USA	8	4–29	cough (7/9); wheezing (3/9); dyspnea (1/9); hemoptysis (2/9); nasal congestion (3/9); recurrent otitis media (OM) (3/9); recurrent fever (4/9); gastrointestinal symptoms (5/9); in 7/8; hematochezia (1/9); FTT (2/9); eosinophilia in 5/8	yes (9/9)	yes (1/9)	NS	yes (6/6)	NS	sIgE (1/3), SPT (1/2)	yes (8/9)	positive challenge (3/3)
10	Sigua, 2013	USA	1	12	persistent cough; progressive anorexia; intermittent fever; weight loss; iron deficiency anemia	yes (CXR)	no	BAL	yes	NS	sIgE –ve	yes	doubtful, based on CH
11	Yavuz, 2014	Turkey	1	36	respiratory distress; hemoptysis; recurrent bronchitis; FTT; iron deficiency anemia; eosinophilia; increased inflammatory index;	yes (CXR, CT)	yes	BAL	NS	NS	sIgE +ve	yes	doubtful, based on CH
12	Mourad, 2015	USA	1	17	severe anemia; respiratory distress	yes (CXR)	yes	BAL	IgG	NS	sIgE +ve	yes	doubtful, based on CH
13	Alsukhon, 2017	USA	1	2	FTT; recurrent diarrhea; persistent cough; tachypnea; high inflammatory markers	yes (CXR)	NS	NS	IgG4	NS	sIgE –ve	yes	NS
14	Ojuawo, 2019	Nigeria	1	4	FTT; cough; dyspnea; wheeze; rhinitis; gastrointestinal symptoms; anemia	yes (CXR)	NS	NS	NS	NS	NS	yes	NS doubtful, based on CH
15	Koc, 2019	Turkey	1	6	massive hemoptysis; hematemesis; deep anemia	yes (CXR, CT)	yes	gastric washing	NS	NS	NS	yes	NS
16	Liu, 2020	China	1	4	respiratory failure; hematochezia; diarrhea; elevated WBC and C-reactive protein	yes (CXR, CT)	no	sputum or fasting gastric fluid	NS	NS	–ve	yes	NS

Abbreviations: BAL, bronchoalveolar lavage; CH, clinical history; CT, computerized tomography; CXR, chest X-Ray; ID, intradermal test; NS, non specified; SPT, skin prick test; –ve, negative; +ve, positive.

**Table 2 nutrients-13-01710-t002:** Differential diagnosis between Heiner syndrome (HS) and idiopathic pulmonary hemosiderosis (IPH).

	HS	IPH
Age	infants or young children	older children/adults
Hemosiderosis	often	always
GI symptoms	often	rarely
Precipitins	yes	no
Response to diet	yes	no
Prognosis	good	variable

**Table 3 nutrients-13-01710-t003:** The real existence of Heiner syndrome: pros and cons.

Pros	Cons
Multiorgan involvement (in particular lung and GI)	Absence of case–control studies
Detection of precipitating antibodies	Precipitating antibodies not pathognomonic
Scarce response to non anti-inflammatory drugs	In most cases the additional administration of anti-inflammatory drugs probably resolved hypersensitivity pneumonia or PH
Clinical improvement after milk removal	The presence of milk in pulmonary infiltrates reported only in one case
Symptoms’ reoccurrence after milk reintroduction	Confirmatory challenge not provided in most cases and/or not adequately performed

## Data Availability

Not applicable.

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
