# Peer review of "Heiner Syndrome and Milk Hypersensitivity: An Updated Overview on the Current Evidence"

_nutrients, 2021, doi:10.3390/nu13051710_

Round 1

Reviewer 1 Report

"Heiner syndrome and milk hypersensitivity: an updated over- 2

view on the current evidence." by S.Arasi et al is an interesting overview on Heiner's syndrome, a rare clinical entity with connection with milk hypersensitivity.

The review is structured in a clear manner and presents known evidence on the syndrome.

The mode of presenting data is interesting and accurate. 

Some further interpretation of Figure 1 could benefit the result.

Author Response

We thank the Reviewer for his/her positive evaluation and useful comment. We have implemented the discussion section on diagnostic criteria described in Figure 1 as suggested (lines 383-386 in the marked manuscript).   

Reviewer 2 Report

In the present manuscript, a narrative, non-systematic review on a rare form of a  respiratory non-IgE mediated CMPA disease ( Heiner syndrome) is presented. The manuscript is interesting and well written, but some changes should be made before being considered for publication

Minor comments: 

Line 302 X-ray (XR) inflitrates should be changed to chest infiltrates on X-ray (XR)

Line 298 refractory should be changed to refractoriness

Line 302 deep should be changed to severe

Line 305 lymphoid should be changed to lymph node

Line 397 salt should be changed to cloride

Line 398 swallow evaluation should be changed to sweat test

Line 398 determined should be changed to ruled out

Line 444 specific should be changed to nor specific

Line 445 duration time of the diet should be changed to extent of the exclusion diet

Line 453 suspect should be changed to suspicion

Major comments

The authors indicated that they have developed a "scoring system" for the disease. In fact, there is no score at all. The authors displayed their own diagnostic criteria for probable and convincing Heiner syndrome, being probable if infiltrates are present in chest XR, and symptoms resolve after milk removal from the diet, and convincing if symptoms recur after milk reintroduction.  This approach is consistent with general non IgE mediated CMPA  diagnostic management. It could be useful, but it is not a score.

The description of the publications included in the review is tedious for the reader. 10/16 of the included papers are just descriptions of a single case. In my opinion, the table is informative enough along with an aggregated summary of the included papers.

Author Response

We are grateful to the Reviewer for his/her useful comments. A detailed point by point reply is provided below. [Q= question, A= answer]

Reviewer’s minor comments: 

Q1- Line 302 Chest infiltrates on X-ray (XR)should be changed to chest infiltrates on X-ray (XR)

Q2- Line 298 refractory should be changed to refractoriness – The text has been changed accordingly

Q3- Line 302 deep should be changed to severe

Q4- Line 305 lymphoid should be changed to lymph node

Q5- Line 397 salt should be changed to cloride

Q6- Line 398 swallow evaluation should be changed to sweat test

Q7- Line 398 determined should be changed to ruled out

Q8- Line 444 specific should be changed to nor specific

Q9- Line 445 duration time of the diet should be changed to extent of the exclusion diet

Q10- Line 453 suspect should be changed to suspicion

Authors’ reply (A1-10)– Thanks again. We have amended the text according to each suggestion.

Reviewer’s major comments

Q11 -  The authors indicated that they have developed a "scoring system" for the disease. In fact, there is no score at all. The authors displayed their own diagnostic criteria for probable and convincing Heiner syndrome, being probable if infiltrates are present in chest XR, and symptoms resolve after milk removal from the diet, and convincing if symptoms recur after milk reintroduction.  This approach is consistent with general non IgE mediated CMPA  diagnostic management. It could be useful, but it is not a score.

A11 – We agree on Reviewer’s consideration. This diagnostic approach is consistent with that one of other non IgE mediated CMPA diseases. However, we think that in such a not-well- known disease as Heiner syndrome our diagnostic approach can help clinicians in their diagnostic pathway. According to the Reviewer’s suggestion, we have changed the terms “diagnostic scoring system” in “diagnostic criteria”

Q12 -The description of the publications included in the review is tedious for the reader. 10/16 of the included papers are just descriptions of a single case. In my opinion, the table is informative enough along with an aggregated summary of the included papers.

A12 – We thank the Reviewer for his/her comment. Although we agree that the readers can find the majority of information in Table 1, we still think that a detailed narrative description of cases can enrich the overall frame of this review. We think that it can help clinicians not only in better understanding the heterogeneity among cases and the need of specific diagnostic criteria but also to take confidence with the diagnostic approach we are proposing. If this is fine with the Reviewer we would prefer to keep the narrative description of each paper.

Round 2

Reviewer 2 Report

The authors have made significant changes in the manuscript. However, they still mantain the extensive description of each case. In my opinion, that section should be summarized.

I forward to the editor the final decision on that point

Author Response

COMMENT: "The authors have made significant changes in the manuscript. However, they still mantain the extensive description of each case. In my opinion, that section should be summarized. I forward to the editor the final decision on that point".   Dear Reviewer, Many thanks for Your revision and positive comments. We appreciate Your time and expertise. We cannot visualize on the platform Editor's suggestion that You have kindly solicited regarding the narrative description of each case. As already stated in the previous point by point reply, we thank You for Your comment. Although we agree that the readers can find the majority of information in Table 1, we still think that a detailed narrative description of cases can enrich the overall frame of this review. We think that it can help clinicians not only in better understanding the heterogeneity among cases and the need of specific diagnostic criteria but also to take confidence with the diagnostic approach we are proposing. If this is fine with the Reviewer and the Editor we would prefer to keep the narrative description of each paper. We are looking forward further instructions.